# The role of traditional restaurants in tourist destination loyalty

**Ricardo David Hernández-Rojas** [1☺¤a]*, **Nuria Huete Alcocer** [2☺¤b]

**1** Department of Agricultural Economics, Finance, and Accounting, University of Cordoba, Córdoba, Spain,
**2** Department of Spanish and International Economics, Econometrics and History and Economic Institutions, University of Castilla-La Mancha, Albacete, Spain

☺ These authors contributed equally to this work.
¤a Current address: Avda. Medina Azahara, Córdoba, Spain
¤b Current address: Plaza de la Universidad, Albacete, Spain
* ricardo.hernandez@uco.es

**Data Availability Statement:** All relevant data are within the paper and its Supporting Information files.

**Funding:** The authors received no specific funding for this work.

## Abstract

The aim of this study is to examine the effect that visitor satisfaction with traditional restaurants has on perceptions of the local gastronomy, the overall image of a city and loyalty to that destination. Fieldwork has been carried out in Córdoba, a city in southern Spain famous for being a UNESCO World Heritage city and for its traditional gastronomy. The methodology used is based on structural equation modeling (PLS-SEM). This paper makes a novel contribution in that no previous studies to date have explored satisfaction with traditional restaurants, with respect to the food, the service and the atmosphere. To achieve the proposed objective, a structured questionnaire has been used to find out the opinions of diners in renowned restaurants that base their cuisine on traditional dishes made with quality local ingredients. The results obtained confirm that a satisfactory experience with the food of a traditional restaurant has a positive effect on the image of the destination and the gastronomy of the place, as well as on visitors' intentions to recommend and repeat the visit to said destination. Based on the analysis carried out, effective strategies are suggested to help manage these types of restaurants. The study provides theoretical and practical implications from a gastronomic perspective, which can enable tourism managers to employ new strategies to retain tourists visiting a city, based on increasing their post-experience satisfaction with restaurants featuring local cuisine.

## Introduction

Gastronomy has different forms of expression and channels for reaching the consumer. The most common is through local restaurants, where it is consumed by tourists and visitors. From an academic perspective, the different types of restaurants are reflected in the multitude of gastronomy studies addressing them: luxury restaurants [1], sustainable restaurants [2], restaurants in prestigious guides [3], restaurants located in hotels [4], restaurants or street food stalls [5] Halal restaurants [6], or fast food restaurants [7] among others. However, there are fewer studies that refer to traditional restaurants; in other words, restaurants that base their offer on the traditional cuisine of their location. Such restaurants revitalize the production of

**Competing interests:** The authors have declared that no competing interests exist.

local ingredients, the taste of the regional cuisine, the gastronomic tradition and the consumers' dining experience. Authors have shown how traditional restaurants emphasize the importance of local food and attract visitors [8]. Therefore, gastronomic tourism adds value to local products and, according to a study by [9], enhances the quality of the tourist experience and makes it more unique. Other academic studies find that traditional food plays a role in diner satisfaction [10]. The study by [10] was the first to study the ambience, the food and the service of traditional restaurants.

Thus, the literature shows how traditional restaurants play an important role in the creation and dissemination of the gastronomic offer of a place. Traditional gastronomy reflects a culture along with the lifestyle, food, and different cultural practices of the inhabitants of a region [11–13]. In turn, tradition is derived from the culture of the place, the food, art, rituals and experiences of a people. Therefore, in the context of this research, we highlight the importance of traditional gastronomy as part of the culture of a region, highlighting how it is a pull factor attracting tourists to a destination [14]. In the current context, traditional gastronomy is called "new traditional cuisine", because it develops a new traditional cuisine from the old [15]. This encourages and promotes certain tourist typologies such as gastronomic tourism [16]. The relationship between gastronomy and tourism has been discussed by numerous authors [17, 18]; however, there has been less research on the relationship between gastronomy in a tourist destination and satisfaction in restaurants explored in terms of food, service and the ambience. As such, this is one of the main contributions of this article. A destination is enriched if its traditional cuisine has a strong presence through traditional restaurants, or if its cuisine is based on local produce, also referred to as zero kilometer food [19]. In this regard, this research is relevant in highlighting two aspects: first, the relevance of gastronomy due to the implications in terms of visitors repeating and recommending a trip to the destination; second, the implications for the management of the restaurant itself and for the management of the city through the public policies designed by policymakers. Therefore, traditional gastronomy affects two aspects of the destination: its economy and the conservation of its regional culture. In terms of the economic aspect, traditional gastronomy uses local produce, generating synergies in the local economy. Regarding the conservation of culture, some authors highlight the revival of traditional recipes and ancestral customs through traditional gastronomy [20, 21]. Consequently, regional cuisine is becoming a more important focus of research in the field of tourism studies [22]. Regarding the management of the restaurant itself, the food, the service and the atmosphere are three basic factors to study in order to guide the proper management of a restaurant [23–25]. In terms of the tourism management of the city, gastronomic tourism based on traditional cuisine can represent a great opportunity for the development of tourist destinations [26, 27]. Consequently, this study adds to the research on variables such as satisfaction with traditional restaurants, traditional gastronomy, the overall image of a destination and loyalty to a tourist destination, with an application to the city of Córdoba.

Traditional restaurant cuisine can trigger sensory memories of the flavors of family food and can play an important role in the sustainable tourism experience because it appeals to the visitor's desire for authenticity within the experience of visiting a destination. Thus, seek experience with local food, the unique flavors, customs or traditions, along with the quality of food, and local raw materials using concepts and movements such as the so-called Slow Food [28]. "Local food" has the potential to enhance the visitor experience by connecting consumers to the region and its perceived culture and heritage [29], just as a meal made with local produce can generate a sense of place [30]. Therefore, it can be considered as tourism entrepreneurship, a need for survival for hospitality and innovation companies and tourism in general in a city [31]; where the traditional food, as an indispensable part of cultural heritage, can boost tourism and, thanks to it, support the economic development of a place [8, 32]. One example is

traditional Chinese restaurants, which, by offering authentic traditional food to consumers, play a relevant role in promoting the culture and local gastronomic heritage [33]. In this respect, visitors' satisfaction with their experience with local cuisine can have consequences for the overall image of a certain tourist destination, perceptions of its gastronomy and destination loyalty, in terms of the visitors' intention to return to or recommend a place. This study explores these relationships in relation to Córdoba, a city declared a UNESCO World Heritage Site.

Thus, this paper addresses destination loyalty, taking into account satisfaction in traditional restaurants and how this satisfaction affects perceptions of the gastronomy and the image of the tourist destination in general. The research also seeks to determine the impact of gastronomy and how it influences tourists' loyalty when it comes to making a repeat visit or recommending a world heritage city such as Córdoba. The analysis is aimed at answering two main questions: Does satisfaction with the restaurant influence the perception of the gastronomy and the image of the destination? Is loyalty to the destination influenced by restaurant satisfaction? To that end, this study first reviews the conceptualization of satisfaction with traditional restaurants as a central axis of gastronomy. Furthermore, we present the theoretical framework of restaurant food, service and ambience as variables related to gastronomy, as well as the overall image of the destination, and how all this can influence loyalty to a tourist destination. Next, we describe the research methodology used; involving the application of structural equation modeling (SEM) to data obtained through a survey administered to tourists after their gastronomic experience in restaurants renowned for their quality and traditional Cordoba dishes. We then discuss the key findings of the empirical analysis carried out, before presenting the conclusions with respect to the hypotheses raised and research questions asked. The results provide an understanding of the relationships between gastronomy, satisfaction and loyalty stemming from the tourists' visit to the restaurant. Some theoretical and practical management implications are drawn regarding how to harness gastronomy to retain tourists visiting a city, based on increasing their post-experience satisfaction with restaurants featuring local cuisine.

## Literature review

### Background on tourist destination loyalty: Satisfaction with the traditional restaurant

In tourism, the concept of loyalty is understood as a behavioral intention in terms of future plans or desires to visit a place again and to recommend a destination [34]. Developing customer loyalty is an important marketing strategy due to the benefits of retaining existing customers [35]; indeed, it has been estimated that, in any market, winning over a new customer costs five times more than retaining an existing one [36] [8]. Thus, this variable is considered a key indicator of tourism success [37], and has therefore been extensively analyzed by numerous researchers [38]. However, loyalty in the tourism context can be difficult to study due to the relatively high costs for individuals to return to a destination [39–41].

Oppermann (2000) [42] suggested that research on loyalty should emphasize the behavioral approach, as it shows the benefits that a loyal tourist can bring to a tourist destination [43]. As revealed by the literature review, a number of studies focusing on tourist behavior [44, 45] have claimed that the variables most often used to explore visitors' loyalty to a given place are the intention to return to the destination and the intention to recommend it [46]. These tourist intentions can provide information to tourist destination managers on how to improve their image [47]; indeed, some studies have suggested [45, 48] that both the cognitive and affective components of the image affect the behavioral intentions of tourists. However, other studies suggested that the real key to attracting tourists to a destination may depend more on the

general image than on any other more specific dimension [49]. In this vein, [46] confirmed the positive relationship between the overall image and the intention to recommend the destination to others.

Apart from the overall image of the destination, there are other antecedents of loyalty that play a greater role, such as tourist satisfaction [50], which is the central focus of this research and has been widely discussed in the academic literature [44, 45, 51, 52]. One of the clearest definitions of satisfaction was given by [53], when he described satisfaction as an individual's response to a cognitive process where the consumer experience is compared with their expectations. Therefore, most research agrees that destination loyalty is positively influenced by tourists' satisfaction with their own experiences at the destination.

In this context, the gastronomy of a place can be viewed as a key tourist experience [54] [38]. The local cuisine served in traditional restaurants is a symbol of the cultural identity of a place or the accumulated history of a region [55]. Thus, local food, as part of the tourist experience, can promote loyalty to a destination [56]. Its ability to help ensure tourist destination loyalty makes the tourist's gastronomic experience a factor to evaluate. Authors such as [57] suggest that tourists' experience of the service, products and other resources can influence their behavior in terms of in their intention to return to the place or recommend it.

However, there are consumers who criticize traditional restaurants for their old-fashioned products, unfamiliar flavors, poor service and traditional decor that is not explained as such. Thus, remaining competitive in the market is becoming one of the most significant challenges for traditional restaurants [8]. In order to maintain their profitability and ensure their long-term survival, these types of restaurants must focus on how to retain existing consumers and improve their loyalty [58]. To that end, and to assure customers' satisfaction with the traditional restaurant, studies such as that of [8] have suggested improving customer service and paying attention to quality control of attributes relating to the food and the ambience (decoration, cleanliness, music, etc.).

Other studies have also confirmed the relationship between satisfaction with the restaurant and subsequent loyalty to it, with personalized service and food found to be essential factors [59]. Along the same lines, but in Korean restaurants, the findings reported by [60] demonstrated how high levels of satisfaction with the restaurant led to loyalty to the restaurant and to the destination itself. Some authors have highlighted the visit to traditional restaurants as a component of tourist satisfaction [61]. One study of restaurants providing online food delivery services confirmed that loyalty to the restaurant brand was related to satisfaction [62]. Evidence of such a relationship has also been found for fast food restaurants, although it was not studied in relation to a destination [63]. However, there are no studies to date that specifically study the relationship between traditional restaurants and destination loyalty, focusing on variables such as food, service and atmosphere.

Although there is an extensive body of research on European restaurants, there has been less of a focus on studying traditional restaurants; as such, this represents a contribution of the present paper to the literature. The literature review has indicated that satisfaction with the traditional restaurant is a possible factor that may directly affect the loyalty to a tourist destination [60]. However, such satisfaction can also affect loyalty indirectly through other factors such as perceptions of the city's gastronomy in general and the overall image of the destination. Accordingly, we carry out a more extensive literature review to identify the direct effects of satisfaction on other factors.

## Satisfaction with the restaurant and the perception of gastronomy

The study by [64] showed that gastronomy components such as the quality of the food, the quality of the service, the restaurant's atmosphere and the perception of fair prices had a

positive impact on customer satisfaction. The results also showed that of all the factors, the quality of food was the most important factor affecting customer satisfaction, which in turn had a positive impact on customer loyalty. Therefore, the experience in restaurants is linked to the perception of the gastronomy of the region [65]. With regard to variables such as the restaurant service, that is, attending to the diner from the moment they enter the restaurant until they leave, we can highlight numerous studies that break down the factors that determine good service: professional attire, personal hygiene [66–68], attention, speed and timeliness in serving the customer, responsiveness, willingness to help, willingness to answer questions [66, 67, 69, 70], knowledge of the dishes on the menu, knowledge of the ingredients [67, 68], smiles from the front-of-house staff, kindness [66, 68, 71, 72], caring and paying individual attention to customer needs [66–68, 71] and having sufficient communication and language skills to be at the same cultural level as the client [67]. Thus, managing and developing positive relationships with customers is a critical factor in the success of a restaurant. One study [73] demonstrated that asking diners about their satisfaction with the food or service at the end of the meal [74] positively influenced their satisfaction. On the other hand [75], studied customer satisfaction in restaurants in Jordan, but focused on personal and functional aspects of employee behavior and the influence on customer satisfaction [69]. Highlighted the factors of waiting time and quality of service, and their role in ensuring customer satisfaction.

Finally, regarding the restaurant environment, different studies directly relate the restaurant's atmosphere or ambience to customer satisfaction [76]. Therefore, it is a factor that restaurant managers and owners should take into account [23]. There are studies that also include external atmosphere or ambience factors such as atmospheric variables [77].

## Satisfaction with the restaurant and the overall image of a destination

Following Bigné [51], suggested that the perceived image of the destination by the visitor after the trip is shaped by the overall impression acquired. The tourist's satisfaction when consuming or using certain services forms part of that impression. In this regard, tourists' perceived image and satisfaction are fundamental determinants of tourist loyalty to a tourist destination. Some studies report that food can play a role in improving the competitiveness of a destination and making it attractive to visitors [9, 78].

Thus, the effect that tourist satisfaction with local food can have on their perception of the image of a tourist destination is especially important. The research carried out by [79] referring to sporting events showed that satisfaction significantly affects the tourist's attitude and perceived image. Others studies report similar findings but relating to the image in the gastronomic tourism industry in general [54]. However, there are no studies in the literature that address the effect of tourist satisfaction with the traditional restaurant on the perception of the general image of the city; more specifically, the image of a tourist destination declared a World Heritage Site. The attempt to demonstrate and explain said effect is another key contribution of this research.

## Traditional gastronomy and destination loyalty

Gastronomy is a broad concept that includes cooking [80], with cuisine being the sum of the dishes resulting from applying production tools and techniques to raw ingredients and/or intermediate preparations, with the characteristic features of these techniques and ingredients giving rise to specific types of cuisine. In the case of local cuisine, traditional restaurants are defined as those that use certain ancestral techniques [33] and purchase raw ingredients according to the proximity of local food [81]. Consequently, many authors conclude that a traditional restaurant is one that is based on the local food movement and that sources produce

mainly or exclusively from local farmers [49, 82, 83], making recipes that have been passed down from generation to generation [33].

With regard to the food that the diner consumes in restaurants, it can involve different degrees of preparation, from largely unaltered raw ingredients to very complex elaborations. Food and its impact on tourist loyalty has been studied from the perspective of food safety. Some studies [84, 85] concluded that the peace of mind about food gained through healthcare coverage helped generate a positive relationship with satisfaction and loyalty. Another study relating to food found that environmentally responsible behavior was positively influenced by cultural sustainability, and negatively by environmental sustainability [86]. A study of gastronomic festivals highlighted how the gastronomic experiences and atmosphere of the food on offer is related to the event brand, which together with the image of the destination have a positive influence on destination loyalty [87]. However, there are no studies to date that explore the components of the gastronomy construct from the point of view of traditional food along with the service and atmosphere of the restaurant. In conclusion, academic studies point towards a direct relationship between food and loyalty. However, other authors point out that this relationship is conditioned by the tourist's interest in gastronomy [13, 88].

### Overall image and destination loyalty

The image of the destination is widely recognized as a relevant phenomenon that influences tourists' decision-making, choice of destination, post-trip evaluation and future behavior [89–92].

Tourist destinations, with international or national tourist demand, hospitality companies compete to attract more tourists to generate income [16]; but they also compete to ensure repeat visits from tourists, and the more times a tourist visits a destination, the greater the loyalty they will show to it [43]. Accordingly, a fundamental element in securing said loyalty to the destination is the perception (overall image) that the tourist has of their satisfaction with the destination.

Authors such as [45] and [48] claim that both the cognitive and affective components of the image affect tourists' behavioral intentions. However, other studies, such as that of [46], contradicts the one carried out by [93], arguing that the intention to recommend is only affected by overall impressions and not by the different dimensions of the image [46]. Thus, a high quality tourist experience or a good impression affects not only the intention to return, but also the intention to recommend. On the other hand, numerous studies [44, 51, 52, 94] have found that destination image is significantly related to tourists' intention to return to a destination, and make a positive word-of-mouth recommendation. This underscores the importance of tourists' satisfaction with the destination, a fundamental component of which can be their satisfaction with the gastronomy in general, and with traditional restaurants in particular.

### Materials and methods

Repeated visits to and/or recommendations of destinations are important because they have a multiplier effect on the profitability of restaurants [25] and therefore on the products and services of the destination itself. Based on the literature review, and in order to achieve the objective of this research and determine which variables influence tourist destination loyalty (LOYDEST), we propose an analysis that allows us to test, on the one hand, how satisfaction with traditional restaurants (SATREST) affects the general perception of gastronomy (GASTRO) and the overall image of the destination (OVERALLIM); and on the other, how such visitor satisfaction after visiting a traditional restaurant can affect the loyalty to a tourist

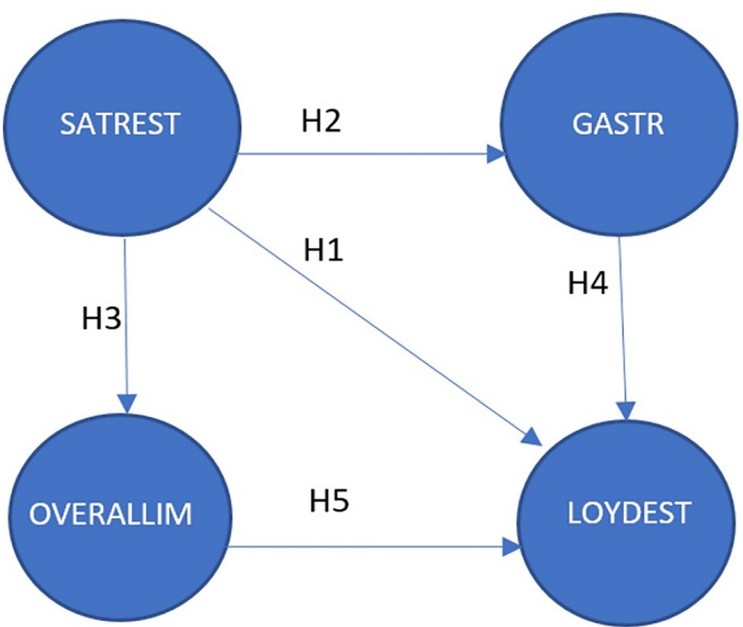

**Fig 1. Proposed research design.** Source: authors.

destination. In this case, the analysis is applied to the city of Córdoba (Spain). Thus, in this section we propose five hypotheses for the empirical research, based on the analyzed literature:

*H1. Satisfaction with the traditional restaurant has a positive and significant influence on destination loyalty.*

*H2. Satisfaction with the traditional restaurant has a positive and significant influence on the perception of the local gastronomy.*

*H3. Satisfaction with the traditional restaurant has a positive and significant influence on the overall image of the destination.*

*H4. The perception of the local gastronomy has a positive and significant influence on destination loyalty.*

*H5. The overall image of the destination has a positive and significant influence on destination loyalty.*

The relationships between the different factors can be seen in proposed model (Fig 1).

## Methodology

### Description of Córdoba and its gastronomy

Córdoba (Spain) is located in the southernmost region of Europe. It is a well-connected inland city with a notable tangible heritage component. As of 2021, it has four UNESCO World Heritage listings and therefore receives a large number of tourists every year. It has an important traditional gastronomy with a long history. Local raw ingredients feature prominently in its traditional dishes, as a consequence of being in a province with an agri-food economy. The different civilizations and cultures that this region has hosted, including the Iberians, Romans, Visigoths, Muslims, Christians and Jews, have all contributed to its unique traditional gastronomy [95]. The region is noted for its quality local produce, as it has seven protected

designations of origin [14]. It is also associated with renowned traditional dishes such as Cordoba salmorejo [96] or oxtail [97], traditional dishes of Arab origin (shoulder of lamb with honey or rosemary; Cod with cinnamon; Mozarabic monkfish) or Sephardic dishes, such as aubergines with honey, together reflecting the legacy of different cultures over the centuries [26]. The mixture of cultures, quality raw ingredients, and recipes handed down over generations all make the traditional gastronomy of Córdoba a point of reference [98]. Restaurants in the tourist area, such as the old part of the city, mainly serve traditional recipes, and the number of restaurants located in this area represents more than 50% of the total in the city [99].

## Questionnaire and scales

The restaurants where the visitor survey was conducted were located in the historic center of the city. This is the most touristic area of the city. The restaurants were selected for their location; secondly, for being recognized by professionals as examples of the traditional gastronomy of Cordoba and finally where the dishes had ingredients with the Protected Designation of Origin of the area.

The information collection procedure was carried out in December 2019 during the weekends, at mealtime, through a questionnaire where the pollster personally asked the tourist about his gastronomic experience. Questionnaire responses were collected by the pollster on the tablet.

The access by the surveyors to the traditional restaurants and the conduct of interviews with tourists following his gastronomic experience was authorized by the managing body and owner of the traditional restaurants. Prior to the completion of the questionnaire, tourists were informed of academic purposes and anonymity in answering. Consent to take the questionnaire was verbal. At all times, the visitor's anonymity to the traditional restaurants was guaranteed.

The validation of the survey and the wording of the questions (in English and Spanish) is based on established items from previous research [100]. The questionnaire was organized into sections. The first includes questions about the tourists' overnight stays and origin. The following sections ask about their satisfaction with the traditional restaurant, the gastronomy in general, the overall image of the tourist destination, as well as their loyalty in terms of their intention to recommend a visit to the city of Córdoba or to visit it again themselves. Regarding the measurement of the variables, the method chosen for this research was the 5-point Likert scale (1 = totally disagree and 5 = totally agree). Previously tested questions from other studies,

**Table 1. Scales used.**

| Authors | Dimension | Indicators |
|---|---|---|
| [8, 10, 64, 74] | Satisfaction with the restaurant (SATREST) | (SATREST1) Renown (known abroad) (SATREST2) Tradition and roots in the local community (SATREST3) Location (SATREST4) Organization (SATREST5) Power of attraction (SATREST6) Food (SATREST7) Service (SATREST8) Atmosphere in the Restaurant |
| [7, 8, 45, 101] | Gastronomy (GASTR) | (GASTR1) Córdoba has exciting gastronomy (GASTR2) Córdoba is gastronomy (GASTR3) Córdoba is cultural heritage (GASTR4) Good value for money in restaurants |
| [45, 48] | Overall Image (OVERALLIM) | (OVERALLIM1) The overall image is positive (OVERALLIM2) It was worth coming to Córdoba (OVERALLIM3) It is a good place to visit (OVERALLIM4) It has a good reputation |
| [44, 45, 51, 52, 101, 102] | Destiny loyalty (LOYDEST) | (LOYDEST1) I will repeat my visit to this restaurant (Cordoba historical center) (LOYDEST2) I will recommend that family and friends come to this restaurant (Cordoba historical center) (LOYDEST3) I will visit Córdoba again (LOYDEST4) I will recommend that family and friends visit Córdoba |

as shown in Table 1, were adapted and used in this research. A pretest of 20 surveys was carried out. Finally, the number of valid questionnaires was 139, yielding a confidence level of 95%.

Thus, the questionnaire is made up of the four parts of the theoretical model (Table 1): restaurant satisfaction, gastronomy, overall image and destination loyalty. After the item refinement process carried out by calculating Cronbach's alpha coefficient for each construct, a total of 20 items were used in our model. Items on the tourist's sociodemographic profile were included at the end of the survey.

The data from this study have been tabulated and analyzed using the IBM SPSS 23 statistical system (IBM Corporation, Armonk, NY, USA) and the SmartPLS 3.3.3 structural equation software package. In the related literature, SEM is considered the most appropriate method to validate the hypotheses proposed in the structural equations and to confirm the complex relationship model. The partial least squares structural equation modeling (PLS-SEM) method applied in this study is a tool used for the analysis of complex interrelationships between observed and latent variables, and has been widely used and validated in tourism research [103].

## Results and discussion

The main findings from the fieldwork are described below, broken down by the blocks addressed in the questionnaire. In the first place, the results of the descriptive analysis regarding the sociodemographic profile of the tourists surveyed during their visit to Córdoba reveals that the majority were women: 57% of those interviewed, compared to 43% who were men. A large share of the respondents were middle-aged people between 40 and 59 years old (50%), and with university studies (55%).

Second, before testing the hypotheses raised in this research, the measurement model was evaluated with PLS [104]. The individual reliability of the items, the reliability of the scale, the convergent validity and the discriminant validity were analyzed. To do so, the Smart PLS 3.3.3 software was used and the significance of the parameters was determined through bootstrapping, which assesses the accuracy of the PLS estimates [105].

### Evaluation of the measurement model: reliability and validity

First, the reliability of the individual indicators was checked (Table 2); the simple correlations of the measures with their respective constructs were analyzed and items with loadings greater than or equal to 0.707 were verified as reliable [106], which means that more than 50% of the variance of the observed variable is shared with the construct.

The composite reliability was then assessed to confirm whether each indicator satisfactorily measures the construct to which it is assigned. This analysis is performed by calculating Cronbach's Alpha [107], the values of which must exceed the cut-off of 0.7, and preferably 0.8. Table 3 shows that all the constructs satisfy the reliability requirement, and three of the four values surpass the more stringent requirement ($> 0.8$), therefore, they validate the internal consistency of the model.

Regarding the validity of the model, as a first step, the convergent validity was assessed (Table 4) by calculating the Average Variance Extracted [91] for each of the constructs. According to [108], this measure indicates the amount of variance that a construct obtains from its indicators relative to the amount of variance due to measurement error. It is recommended that its value be equal to or greater than 0.5, which indicates that each construct explains at least 50% of the variance of the assigned indicators [103, 108, 109]. We also calculated another of the most important reliability measures for PLS; namely, rho_A [110], 2015).

**Table 2. Individual item reliability (Reflective).**

| Factor | Indicator | Loading |
|---|---|---|
| SATREST | SATREST1 | 0.702 |
| | SATREST2 | 0.802 |
| | SATREST3 | 0.801 |
| | SATREST4 | 0.824 |
| | SATREST5 | 0.740 |
| | SATREST6 | 0.835 |
| | SATREST7 | 0.739 |
| | SATREST8 | 0.772 |
| GASTR | GASTR1 | 0.761 |
| | GASTR2 | 0.742 |
| | GASTR3 | 0.728 |
| | GASTR4 | 0.734 |
| OVERALLIM | OVERALLIM1 | 0.781 |
| | OVERALLIM2 | 0.862 |
| | OVERALLIM3 | 0.885 |
| | OVERALLIM4 | 0.740 |
| LOYDEST | LOYDEST1 | 0.719 |
| | LOYDEST2 | 0.833 |
| | LOYDEST3 | 0.819 |
| | LOYDEST4 | 0.831 |

Next, we checked the discriminant validity of the measurement model, which indicates the extent to which a given construct is different from the others. In this case, for the discriminant validity requirement to be fulfilled, the variance shared between a variable and its indicators must be greater than the variance shared with the other variables of the model [104] (Barclay et al., 1995). To evaluate it, this study uses the correlations of the latent variables (Table 5): the diagonal of the matrix shows the value of the square root of the AVE of the corresponding construct. Since these values are greater than the correlations between the constructs, the constructs met the discriminant validity requirement. Additionally, to verify the measurement instrument, we also calculate the HTMT values. As Table 5 shows, the values indicated in parentheses were consistently lower than 0.9 [111].

## Evaluation of the structural model

Having validated the measurement model, the structural model was evaluated. Table 6 shows the path coefficients (β) indicating the relationships between the constructs, as well as the significance of these relationships. To study the stability and significance of the estimated parameters, the abovementioned non-parametric resampling technique of bootstrapping was used, which consists of repeated random sampling with replacement of the original sample to create a number of bootstrap samples [105].

**Table 3. Composite reliability.**

| Factor | Cronbach's Alpha | *Composite reliability* |
|---|---|---|
| **SATREST** | 0.906 | 0.924 |
| **GASTR** | 0.728 | 0.830 |
| **OVERALLIM** | 0.834 | 0.890 |
| **LOYDEST** | 0.814 | 0.878 |

**Table 4. Convergent validity.**

| Factor | AVE | Rho_A |
|---|---|---|
| **SATREST** | 0.606 | 0.909 |
| **GASTR** | 0.549 | 0.728 |
| **OVERALLIM** | 0.671 | 0.836 |
| **LOYDEST** | 0.643 | 0.821 |

Furthermore, in order to assess the structural model, we calculated $R^2$ to analyze whether a substantial part of the variance of the dependent variables is explained. According to [112], should be greater than or equal to 0.1. Table 7 shows the results obtained, greater than 0.1, confirming the predictive relevance of the model studied.

Thus, after the analysis of the structural model, all the hypotheses formulated are confirmed. The first hypothesis (H1) posited that satisfaction with the traditional restaurant positively and significantly affects the loyalty to a tourist destination such as Córdoba. In other words, when a visit is anticipated, whether due to a recommendation or a return visit to a certain restaurant, the loyalty directed towards the restaurant is transferred to the destination. Studies of heritage cities have identified the same relationship with the heritage elements [113]. In the same vein, research by [60] demonstrated how a high level of satisfaction with the traditional restaurant, in this case Korean restaurants, led to loyalty towards the restaurant and towards the destination itself, reflected in the intention to return to the city or recommend it. In terms of practical implications, city tourism managers should ensure a minimum quality in restaurants to ensure that they satisfy tourists and visitors, and thereby secure their loyalty. The results indicate that satisfaction has a positive influence on destination loyalty, which fosters tourists' intention to return to the destination in the future, and to recommend it once they return to their place of origin.

Regarding the second hypothesis (H2), it has been shown that satisfaction with the traditional restaurant has a significant and positive relationship with perceptions of the local gastronomy. That is, when the diner, following his/her gastronomic experience, is satisfied with individual factors such as food, service, atmosphere, location, reputation, together with specific features typical of the traditional regional gastronomy, he/she assigns greater value to the gastronomy of the region. These results are consistent with studies such as that by [10] focused on mountain restaurants; by [114] examining fast food restaurants; or by Min [115] exploring ethnic restaurants. If we apply this finding to the management of restaurants, it can be seen as a wake-up call to their managers. The basis of any gastronomic experience is customer satisfaction; therefore, the food itself is not enough. Factors such as the service, the organization, and the ambience all help to increase customer satisfaction with the restaurant; therefore, must take any necessary actions to improve these elements. An example would be innovation in service or product, as they have a positive relationship with customer satisfaction [116].

Third, it has been shown here that satisfaction with the traditional restaurant positively and significantly influences the overall image of the destination (H3), in the Córdoba case study.

**Table 5. Discriminant validity (Fornell-Larcker).**

| | SATREST | GASTR | OVERALLIM | LOYDEST |
|---|---|---|---|---|
| **SATREST** | **0.778** | | | |
| **GASTR** | 0.594 (0.723) | **0.741** | | |
| **OVERALLIM** | 0.639 (0.728) | 0.625 (0.788) | **0.819** | |
| **LOYDEST** | 0.622 (0.730) | 0.602 (0.767) | 0.652 (0.779) | **0.802** |

**Table 6. Test of the hypotheses.**

| Hypothesis | Structural Relationship | Standardized Path Coefficient (β) | t-value Bootstrap | Result |
|---|---|---|---|---|
| H1 | SATREST -> LOYDEST | 0.271 | 3.075** | ACCEPTED |
| H2 | SATREST -> GASTR | 0.594 | 8.689*** | ACCEPTED |
| H3 | SATREST -> OVERALLIM | 0.639 | 10.417 *** | ACCEPTED |
| H4 | GASTR -> LOYDEST | 0.233 | 2.509** | ACCEPTED |
| H5 | OVERALLIM -> LOYDEST | 0.334 | 3.664*** | ACCEPTED |

* $p < .05$

** $p < .01$

***$p < .001$ (based on a one-tailed student's t distribution) t (0.05;) = 1,645; t(0.01) = 2,327; t(0.001) = 3,092.

Of the five hypotheses tested, it is the third one that shows the strongest relationship between satisfaction with the restaurant and the factor under analysis—in this case, the image of a destination. This finding represents a novel contribution to literature. That is, when the visitor to the destination experiences satisfaction with the traditional restaurant, the destination benefits, since the visitor transfers that satisfaction to the perception of the destination.

This is consistent with results from other studies [9, 78, 117]. These authors state that positive experiences with local foods and the activities surrounding them significantly influence the probability of revisiting the place. In terms of practical implications, it is worth promoting different activities after the gastronomic experience since the tourist is more proactive and sees the city differently after a satisfaction experience in the restaurant.

With the fourth hypothesis (H4), we have found evidence that the gastronomy of the region has a positive relationship with loyalty to the destination. That is, when the gastronomy is viewed positively, repeat visits to and recommendations of that tourist destination become more likely. Studies in this line corroborate this finding [87, 118]. Therefore, a bad experience in a traditional restaurant can affect the entire gastronomic image of a region. The importance of striving to ensure tourists have a positive experience in traditional cuisine restaurants are therefore clear.

And finally, with the fifth hypothesis (H5) the positive and significant influence that the overall image of the destination has on destination loyalty is supported. This is consistent with results from other studies such as [44, 51, 52], who verified the direct and indirect relationship between image and loyalty. This reinforces the suggestion that restaurant managers should make an effort to assess the satisfaction of diners, since it directly influences the attitude or desire to repeat the visit to the restaurant and by extension to the destination itself. It is also clear that managers of traditional restaurants need to carry out an objective evaluation of the final result of the gastronomic experience in order to make improvements, since it can end up influencing the general perception that tourists have of the city.

Thus, the results obtained and the discussion of those results highlight the importance of the variables evaluated when designing and implementing strategies that seeks to enhance satisfaction with the traditional gastronomic experience in the destination. Fig 2 below depicts the SEM model obtained:

**Table 7. Predictive relevance of the model.**

| Factor | $R^2$ | $R^2$ TIGHT |
|---|---|---|
| GASTR | 0.353 | 0.348 |
| OVERALLIM | 0.408 | 0.404 |
| LOYDEST | 0.527 | 0.516 |

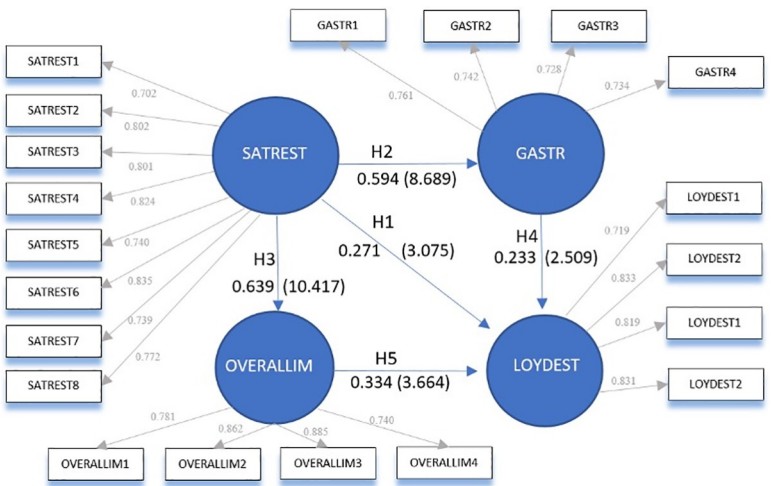

**Fig 2. Casual relationships of the model.** Source: authors.

## Conclusions and limitations

Tourists' gastronomic experience in the restaurants of a destination contributes positively to tourist loyalty, provided that they have a degree of satisfaction with the traditional gastronomic experience. Based on the data obtained, it can be affirmed that satisfaction with the traditional restaurant has a positive influence on perceptions of gastronomy, promoting and creating a gastronomic brand and image of the destination that prompts tourists to revisit and recommend the destination in question. Therefore, tourist and visitor loyalty can be achieved through the satisfaction that the visitor gains from traditional restaurants. Among the factors to achieve satisfaction with the traditional restaurant you must have a minimum of quality in the food, the service of the restaurant next to the atmosphere since getting this quality helps to increase the satisfaction of the visitor or tourist. In addition, it not only influences the likelihood of recommending the destination, but also the overall image of the city of Córdoba and its gastronomy. Therefore, it is essential to implement initiatives aimed at the professionalization of the traditional restaurant. Improvements can be achieved through training, among other actions. Regarding, the initiatives to be implemented, the training should focus on the three main factors: food, service and the atmosphere created by the restaurant. Other possible improvements that may increase satisfaction with the restaurant include engaging the tourist, providing an explanation of the dishes, or improving the organization of the restaurant, all factors that help to a create a positive gastronomic experience.

As the results have shown, there are certain unique values that shape the tourist's perceptions after visiting a traditional restaurant. These include the welcome to the restaurant by the staff, the explanations given of certain traditional dishes, along with the friendliness of the staff and quality of service. In terms of the atmosphere and the organization, the décor is particularly important, sharing a common theme with the traditional dishes. The food in itself is key: the flavor, the traditional textures and the recognition that the dish being sampled is connected to the local territory were the most significant responses. Traditional restaurants thus create a link between the tourist and the destination, positively influencing loyalty to the city. The brand built by certain traditional restaurants, together with the quality of the experience, is an attribute recognized by those travelers who wish to spend part of their trip learning about the local culture through the existing heritage.

Therefore, the managers of the destinations and the managers of the traditional restaurants in the region share the objective of achieving visitor satisfaction. Accordingly, they must carry out joint actions aimed at this goal. The main benefits to be obtained are the loyalty of the visitor and the improvement of both the gastronomy and the overall experience. Thus, visitor loyalty can be clearly influenced by their enjoyment of the local gastronomy, through the restaurants that use a wide range of high-quality local produce cooked in the traditional way or incorporated into different types of food to create innovative yet traditional cuisine. This study highlights how the satisfaction generated by restaurants has become a fundamental pillar in attracting and retaining visitors. For practical purposes, this means that city managers (public administrations) must ensure a minimum quality in restaurants so that they provide satisfaction to tourists. In turn, owners and managers of traditional restaurants in Córdoba have a responsibility to guarantee customer loyalty. As in the previous context, this will also be a great help in the current situation as the COVID-19 pandemic abates, since ensuring a high level of satisfaction with the gastronomic experience is the starting point on the path to recovery, as it boosts competitiveness against any competing destination. Meanwhile, even in times of pandemic, it is difficult to maintain that satisfaction with service and quality, as they are sometimes distorted by the fact that some of the restaurants currently serve take-out food by ordering third-party delivery services. This has been investigated [119] for restaurants generically. However, it can be considered as a future line of research in the coming months for traditional restaurants: the change that customers may experience with these types of restaurants due to the COVID-19 pandemic.

Regarding the limitations of this study, it should be noted that other variables could have been included to explain satisfaction with the restaurant, such as the bathroom facilities, greetings and farewells, the menu and the emotional components. These variables open up avenues of ongoing research. Future lines of study could also focus on the intersection of information between supply and demand in order to provide information on an appropriate balance in specific markets. The proposed model can also be applied to other places with a tourism offer similar to that of Córdoba, which would allow for useful comparisons and the identification of critical elements that can help continually improve customer satisfaction and loyalty at the traditional restaurant.

## Supporting information

**S1 Data.**
(XLSX)

## Author Contributions

**Conceptualization:** Ricardo David Hernández-Rojas.

**Formal analysis:** Nuria Huete Alcocer.

**Investigation:** Ricardo David Hernández-Rojas, Nuria Huete Alcocer.

**Methodology:** Nuria Huete Alcocer.

**Software:** Nuria Huete Alcocer.

**Supervision:** Ricardo David Hernández-Rojas.

**Writing – original draft:** Ricardo David Hernández-Rojas.

**Writing – review & editing:** Ricardo David Hernández-Rojas, Nuria Huete Alcocer.

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
