## [Decision Letter · Decision Letter 0]

21 Apr 2021

PONE-D-21-07254

The role of traditional restaurants in tourist destination loyalty

PLOS ONE

Dear Authors,

Thank you for submitting your manuscript to PLOS ONE. After careful consideration, we feel that it has merit but does not fully meet PLOS ONE’s publication criteria as it currently stands. Therefore, we invite you to submit a revised version of the manuscript that addresses the points raised during the review process.

We look forward to receiving your revised manuscript.

Kind regards,

Dejan Dragan, PhD

Academic Editor

PLOS ONE

Additional Editor Comments:

The paper was reviewed by several reviewers. Their reviews are very diverse, from the major revision all the way to the minor revision of the paper. Therefore, since the average review implies the major revision, the AE has decided a decision "Major revision". Please, carefully follow all the comments of all reviewers. If the improvement of the paper is going to be significant, then it might be a chance to be accepted. AE DD

Journal Requirements:

 [NO].

[NO].

5. Please amend your list of authors on the manuscript to ensure that each author is linked to an affiliation. We note that affiliation 2 is not linked. Authors’ affiliations should reflect the institution where the work was done (if authors moved subsequently, you can also list the new affiliation stating “current affiliation:….” as necessary).

6. We note you have included a table to which you do not refer in the text of your manuscript. Please ensure that you refer to Table 7 in your text; if accepted, production will need this reference to link the reader to the Table.

Reviewers' comments:

Reviewer's Responses to Questions

**Comments to the Author**

1. Is the manuscript technically sound, and do the data support the conclusions?

Reviewer #1: Partly

Reviewer #2: Yes

Reviewer #3: Yes

Reviewer #4: Partly

2. Has the statistical analysis been performed appropriately and rigorously? 

Reviewer #1: Yes

Reviewer #2: Yes

Reviewer #3: Yes

Reviewer #4: Yes

3. Have the authors made all data underlying the findings in their manuscript fully available?

Reviewer #1: Yes

Reviewer #2: Yes

Reviewer #3: Yes

Reviewer #4: Yes

4. Is the manuscript presented in an intelligible fashion and written in standard English?

Reviewer #1: No

Reviewer #2: Yes

Reviewer #3: Yes

Reviewer #4: No

5. Review Comments to the Author

Reviewer #1: Comment #: The paper "The role of traditional restaurants in tourist destination loyalty" is interesting for journal readers. Kindly take note of the following specific comments to make it better.

Paragraphing should be improved. Around 250 words per paragraph should convey a clear message.

# The authors can elaborate more on the contributions of this paper and clear discussion of retults

Dear Author/s

Author can utilize some of the information from the following materials on tourism demand of more recent literature will make the work more relevant to readers.

You will also need to include recent developments to the paper, i.e. covid19.

Need clear future recommendation in the context of innovative and entrepreneurship

Consider adding the following recent papers to the revised work.

• https://doi.org/10.1080/13683500.2020.1816929

• https://doi.org/10.1080/15567249.2016.1263251

• https://doi.org/10.1002/jtr.2151

• https://doi.org/10.1016/j.tourman.2019.01.014

• https://doi.org/10.1177/1354816619888346

• Isik, C. (2012). The USA’s international travel demand and economic growth in Turkey: A causality analysis:(1990–2008). Tourismos: An International Multidisciplinary Journal of Tourism, 7(1), 235-252.

• Işık, C , Günlü Küçükaltan, E , Taş, S , Akoğul, E , Uyrun, A , Hajiyeva, T , Turan, B , Dırbo, A , Bayraktaroğlu, E . (2019). Tourism and innovation: A literature review . Journal of Ekonomi , 1 (2) , 98-154 . Retrieved from https://dergipark.org.tr/tr/pub/ekonomi/issue/50958/669185

• Işık, C , Günlü Küçükaltan, E , Kaygalak Çelebi̇, S , Çalkın, Ö , Enser, İ , Çeli̇k, A . (2019). Tourism and entrepreneurship: A literature review . Journal of Ekonomi , 1 (1) , 1-27 . Retrieved from https://dergipark.org.tr/tr/pub/ekonomi/issue/45934/579359

An argument for the inclusion of the other variables should be mentioned briefly and why their selection.

It is vital that this manuscript is proofread by a native speaker of English Language to further strengthen easy readership.

The authors can also show how this study differs from other studies published in the PLOSONE’s journal.

Will be looking forward to seeing your reviewed manuscript.

Reviewer #2: 1. The definition of traditional restaurant is broad. In this paper, traditional restaurant should be the "new traditional cuisine" formed by the fusion of different cultures over the centuries. While this will lead to a problem: foreign tourists are satisfied because the dishes have the characteristics of their hometown, and then generate destination loyalty, rather than the loyalty brought by the true traditional cuisine of Cordoba. Even if the first aspect of the questionnaire "satisfaction with traditional restaurants" includes an indication of "local", it may also be the satisfaction brought about by the reputation of the local cuisine that includes other regional cultures.

2. In the questionnaire, whether the two instructions in the fourth part "Tourist Destination Loyalty" "I will go to this restaurant again" and "I will recommend this restaurant to my friends and family" should belong to the first part "satisfaction with traditional restaurants". I, my friends and families may have come to Cordoba for conferences or other reasons, so I tasted the restaurant, but not for the purpose of traveling, so this should not be attributed to "tourism destination loyalty" Category.

3 The economic model is simple and the conclusion is obvious

Reviewer #3: The article presents the results of original research concerning the issue of visitor satisfaction with traditional restaurants, perception of local gastronomy, the overall image of a destination, and destination loyalty. The issue has been well positioned with respect to the abundant and current subject-related literature while the obtained data confirmed the hypotheses posed within the study. The paper's strong sides include an important research issue, a well-written literature review, utilization of methodology based on structural equation modeling, presentation of statistical analysis results, key conclusions. The article may provide significant input into the promotion of scientific debate regarding local restaurants and their role in the shaping of cities' images as well as tourist destination loyalty. The methodology used by the authors can be applied by other researchers for conducting comparative studies.

However, the article's text requires additional supplementation which will strengthen the research. The authors should provide more information regarding the obtaining of source material on the basis of which further analyses were performed (such as the number of restaurants where studies were carried out, assumptions upon which restaurants were qualified as traditional).

Conclusions should stipulate the article's scientific contribution. Consideration should be given to the practical implications of the first hypothesis (H1) concerning minimum quality in restaurants (Will minimum quality ensure the satisfaction of tourists?).

To facilitate the carrying out of comparative studies the inclusion of a questionnaire from survey studies within the appendix would be desirable.

The the work would require minor corrections including the removal of duplicated annotations on pg. 16 and pg. 27 (according to other guidelines); correct the numbers of indicators for Gastronomy in Table 1 on pg. 22.

Reviewer #4: 1. The main claims of the paper are very interesting and important for sustainable tourism development. The expansion of culinary tourism based on the traditional cuisine of a given country is perceived as a trend by UNWTO or Slow Food movement/organisation (which could be mentioned in the article).

2. The introduction should draw more attention, describe the problem more clearly and provide rationale for the study. This part of the text seems chaotic, a few issues are repeated.

3. The style should be improved, the text contains many repetitions. The numbers instead of the authors' names also make a bad impression.

4. The survey methodology should be described in more detail. The authors state that the research was conducted using a questionnaire and personal interview but the description of the results shows that only the survey data was taken into account. What was the interview about? Likewise, the description of the course of research is unclear.

5. The results were based on the responses of 139 respondents. It seems that this number is not sufficient to generalize the conclusions and apply them to populations other than tourists visiting traditional retaurants in Cordoba.

6. PLOS authors have the option to publish the peer review history of their article (what does this mean?). If published, this will include your full peer review and any attached files.

Reviewer #1: No

Reviewer #2: No

Reviewer #3: **Yes: **Halina Kiryluk

Reviewer #4: No

---

## [Author Response · Author response to Decision Letter 0]

10 May 2021

Manuscript PONE-D-21-07254, formerly titled “The role of traditional restaurants in tourist destination loyalty”. 

Dear editor 

Thank you very much for allowing us revising and resubmitting tour article to the International Journal PLOS ONE. We have found your comments to be highly helpful in improving the article in terms of the introduction, the theoretical background, the research methodology, and the theoretical contributions. After carefully reading your comments, we have introduced some changes in the manuscript to address your concerns. We have linked the author's affiliation 2. Following your recommendation, as shown in the new version of the manuscript. We present our detailed comments below. 

We are considering that this authentic work is reaching the standards of your journal and we are aware that the rewires comments and suggestions will improve our work.

Thank you once again for you kind collaboration. 

Yours faithfully,

The authors

Dear Reviewer,

We are very pleased to submit a revised version of the manuscript “The role of traditional restaurants in tourist destination loyalty” to the International Journal PLOS ONE. We thank you very much for giving us the opportunity to revise.

We have made every effort to implement the reviewers’ suggestions and thus improve our research. The revision has addressed three main issues:

• We have improved the introduction, literature review, methodology, empirical results, discussion, conclusions, future research and limitations. The article has been rewritten.

• We have provided a rationale for the inclusion of certain parts of the text. We have rewritten certain sentences that were not clear in the previous version and, in response to the reviewers' suggestions, explained the rationale for the inclusion of others.

• The text has been edited by a native English proofreader.

We provide below our detailed responses to your comments. 

Thank you once again for you kind collaboration. 

Yours faithfully,

The authors

RESPONSES TO REVIEWER 1

Dear Author/s

1. Paragraphing should be improved. Around 250 words per paragraph should convey a clear message.

Thank you for your comments. They have helped us improve the manuscript in several ways. Relevant changes in the manuscript are marked in red.

We have improved the introduction, literature review, methodology, empirical results, discussion, conclusions, future research and limitations. The article has been rewritten.

2. Author can utilize some of the information from the following materials on tourism demand of more recent literature will make the work more relevant to readers. You will also need to include recent developments to the paper, i.e. covid19. Need clear future recommendation in the context of innovative and entrepreneurship. Consider adding the following recent papers to the revised work.

• https://doi.org/10.1080/13683500.2020.1816929

• https://doi.org/10.1080/15567249.2016.1263251

• https://doi.org/10.1002/jtr.2151

• https://doi.org/10.1016/j.tourman.2019.01.014

• https://doi.org/10.1177/1354816619888346

• Isik, C. (2012). The USA’s international travel demand and economic growth in Turkey: A causality analysis:(1990–2008). Tourismos: An International Multidisciplinary Journal of Tourism, 7(1), 235-252.

• Işık, C , Günlü Küçükaltan, E , Taş, S , Akoğul, E , Uyrun, A , Hajiyeva, T , Turan, B , Dırbo, A , Bayraktaroğlu, E . (2019). Tourism and innovation: A literature review . Journal of Ekonomi , 1 (2) , 98-154 . Retrieved from https://dergipark.org.tr/tr/pub/ekonomi/issue/50958/669185

• Işık, C , Günlü Küçükaltan, E , Kaygalak Çelebi̇, S , Çalkın, Ö , Enser, İ , Çeli̇k, A. (2019). Tourism and entrepreneurship: A literature review . Journal of Ekonomi , 1 (1) , 1-27 . Retrieved from https://dergipark.org.tr/tr/pub/ekonomi/issue/45934/579359

Thank you very much for your suggestions. They have helped us improve the manuscript in several ways. Relevant changes in the manuscript are marked in red. We reviewed and references studies such as:

[116] Cem, I., et al., Tourism and innovation: A literature review. Journal of Ekonomi, 2019. 1(2): p. 98-154.

[31] Cem Işık, Günlü Küçükaltan, E , Kaygalak Çelebi, S , Çalkın, Ö , Enser, İ , Çelik, A . Turizm ve Girişimcilik Alanında Yapılmış Çalışmaların Bibliyometrik Analizi . . Güncel Turizm Araştırmaları Dergisi 2019. 3 (1) p. 119-149.

[32] Cem, I., The USA’s international travel demand and economic growth in Turkey: A causality analysis:(1990–2008). Tourismos: An International Multidisciplinary Journal of Tourism, 2012. 7(1): p. 235-252.

[16] Dogru, T., Cem Isik, and E. Sirakaya-Turk, The balance of trade and exchange rates: Theory and contemporary evidence from tourism. Tourism Management, 2019. 74: p. 12-23

(See page 3, lines 8-10): 

“[…] how it is a pull pull factor attracting tourists to a destination [14]. In the current context, traditional gastronomy is called “new traditional cuisine”, because it develops a new traditional cuisine from the old [15]. This encourages and promotes certain tourist typologies such as gastronomic tourism [16]. 

(See page 4, lines 14-16; 19-22):

 “[…] it appeals to the visitor's desire for authenticity within the experience of visiting a destination. Thus, seek experience with local food, the unique flavors, customs or traditions, along with the quality of food, and local raw materials using concepts and movements such as the so-called Slow Food [28]. "Local food" has the potential to enhance the visitor experience by connecting consumers to the region and its perceived culture and heritage [26], just as a meal made with local produce can generate a sense of place [27]. Therefore, it can be considered as tourism entrepreneurship, a need for survival for hospitality and innovation companies and tourism in general in a city [31]; where the traditional food, as an indispensable part of cultural heritage, can boost tourism and, thanks to it, support the economic development of a place [8] [32]. 

(See page 12, lines 7-8):

“Tourist destinations, with international or national tourist demand, hospitality companies compete to attract more tourists to generate income [16]; but they also compete to ensure […]”.

(See page 22, lines 11-12):

“[…] necessary actions to improve these elements An example would be innovation in service or product, as they have a positive relationship with customer satisfaction [116].

You will also need to include recent developments to the paper, i.e. covid19. 

Thank you very much for your suggestions. They have helped us improve the manuscript. We have added the following paragraph in the conclusions (see page 26, lines 14-20):

“Meanwhile, even in times of pandemic, it is difficult to maintain that satisfaction with service and quality, as they are sometimes distorted by the fact that some of the restaurants currently serve take-out food by ordering third-party delivery services. This has been investigated [119] for restaurants generically. However, it can be considered as a future line of research in the coming months for traditional restaurants: the change that customers may experience with these types of restaurants due to the COVID-19 pandemic”

3. An argument for the inclusion of the other variables should be mentioned briefly and why their selection.

Thank you very much for your suggestions. They have helped us improve the manuscript in several ways. Relevant changes in the manuscript are marked in red.

(See page 24, lines 9-12):

“[…] satisfaction that the visitor gains from traditional restaurants. Among the factors to achieve satisfaction with the traditional restaurant you must have a minimum of quality in the food, the service of the restaurant next to the atmosphere since getting this quality helps to increase the satisfaction of the visitor or tourist.”

That is why it has been tended as the main variable of this research: satisfaction with the traditional restaurant. Variable that affects image, gastronomy and loyalty.Se podría haber tenido encuentra otras variables como la motivación, la experiencia, etc. But our goal was to analyze a simpler model.

4. It is vital that this manuscript is proofread by a native speaker of English Language to further strengthen easy readership.

The text has been edited by a native English proofreader.

5. The authors can also show how this study differs from other studies published in the PLOSONE’s journal.

This research has been considered relevant to literature in Marketing and Tourism. Research provides a novel part that can be taken into account in these fields. This particular theme of traditional local gastronomy and loyalty has not been published in other works of the journal PLOSONE.

RESPONSE TO REVIEWER 2

Dear Author(s), 

1. The definition of traditional restaurant is broad. In this paper, traditional restaurant should be the "new traditional cuisine" formed by the fusion of different cultures over the centuries. While this will lead to a problem: foreign tourists are satisfied because the dishes have the characteristics of their hometown, and then generate destination loyalty, rather than the loyalty brought by the true traditional cuisine of Cordoba. Even if the first aspect of the questionnaire "satisfaction with traditional restaurants" includes an indication of "local", it may also be the satisfaction brought about by the reputation of the local cuisine that includes other regional cultures.

Thank you very much for the comment. They have helped us improve the manuscript in several ways. Relevant changes in the manuscript are marked in red. We have added a sentence in the introduction (see page 3, lines 7-10):

“[…] how it is a pull factor attracting tourists to a destination [14]. In the current context, traditional gastronomy is called “new traditional cuisine”, because it develops a new traditional cuisine from the old [15.]”

[15] Savitri, A.I., Introducing New Traditional Cuisine for Maintaining Culture and Promoting Tourism in Tegal Regency. Culturalistics: Journal of Cultural, Literary, and Linguistic Studies, 2019. 3(2): p. 7-12.

2. In the questionnaire, whether the two instructions in the fourth part "Tourist Destination Loyalty" "I will go to this restaurant again" and "I will recommend this restaurant to my friends and family" should belong to the first part "satisfaction with traditional restaurants". I, my friends and families may have come to Cordoba for conferences or other reasons, so I tasted the restaurant, but not for the purpose of traveling, so this should not be attributed to "tourism destination loyalty" Category.

We highly appreciate the feedback. We have added a paragraph in order to clarify that traditional restaurants are located in the historic center of Cordoba. This explains why in the questionnaire we have placed the questions "I will recommend this restaurant to my friends and family" should belong to the first part "satisfaction with traditional restaurants", in the section “Tourist Destination Loyalty"

In addition to these two items LOYDEST1: I will repeat my visit to this restaurant (Cordoba historical center) and LOYDEST2: I will recommend that family and friends come to this restaurant (Cordoba historical center) que miden la lealtad a Córdoba, that measure loyalty to Cordoba, was reason to investigate whether when the tourist recommended this city to family and friends, he could also make comments about the good gastronomy of the city "I recommend visiting the city of Cordoba and when you do enjoy traditional restaurants." Likewise, in case they repeated the visit to this city again, if they would enjoy traditional cuisine again. In short we wanted to see if a certain loyalty has been created to the city thanks to traditional cuisine.

Paragraph added: A last paragraph is included in " Methodology, Questionnaire and scales " where questions about restaurant selection and location are answered (see page 15, lines 5-13):

“The restaurants where the visitor survey was conducted were located in the historic center of the city. This is the most touristic area of the city. The restaurants were selected for their location; secondly, for being recognized by professionals as examples of the traditional gastronomy of Cordoba and finally where the dishes had ingredients with the Protected Designation of Origin of the area.

The information collection procedure was carried out in December 2019 during the weekends, at mealtime, through a questionnaire where the pollster personally asked the tourist about his gastronomic experience. Questionnaire responses were collected by the pollster on the Tablet”.

RESPONSE TO REVIEWER 3

1. The article presents the results of original research concerning the issue of visitor satisfaction with traditional restaurants, perception of local gastronomy, the overall image of a destination, and destination loyalty. The issue has been well positioned with respect to the abundant and current subject-related literature while the obtained data confirmed the hypotheses posed within the study. The paper's strong sides include an important research issue, a well-written literature review, utilization of methodology based on structural equation modeling, presentation of statistical analysis results, key conclusions. The article may provide significant input into the promotion of scientific debate regarding local restaurants and their role in the shaping of cities' images as well as tourist destination loyalty. The methodology used by the authors can be applied by other researchers for conducting comparative studies.

However, the article's text requires additional supplementation which will strengthen the research. The authors should provide more information regarding the obtaining of source material on the basis of which further analyses were performed (such as the number of restaurants where studies were carried out, assumptions upon which restaurants were qualified as traditional).

Second paragraph added: A last paragraph is included in " Methodology, Questionnaire and scales " where questions about restaurant selection and location are answered (see page 15, lines 5-9):

“The restaurants where the visitor survey was conducted were located in the historic center of the city. This is the most touristic area of the city. The restaurants were selected for their location; secondly, for being recognized by professionals as examples of the traditional gastronomy of Cordoba and finally where the dishes had ingredients with the Protected Designation of Origin of the area."

2. Conclusions should stipulate the article's scientific contribution. Consideration should be given to the practical implications of the first hypothesis (H1) concerning minimum quality in restaurants (Will minimum quality ensure the satisfaction of tourists?).

Thank you for your comments. They have helped us improve the manuscript (conclusions) in several ways. Relevant changes in the manuscript are marked in red.

(See page 24, lines 9-12):

 “[…]gains from traditional restaurants. Among the factors to achieve satisfaction with the traditional restaurant you must have a minimum of quality in the food, the service of the restaurant next to the atmosphere since getting this quality helps to increase the satisfaction of the visitor or tourist. In addition,”

3. To facilitate the carrying out of comparative studies the inclusion of a questionnaire from survey studies within the appendix would be desirable.

If we need to send the complete questionnaire at the request of the magazine we do it. But we wouldn't like to include it in the manuscript.

4. The work would require minor corrections including the removal of duplicated annotations on pg. 16 and pg. 27 (according to other guidelines); correct the numbers of indicators for Gastronomy in Table 1 on pg. 16.

We highly appreciate the feedback; it has led us to make modifications to improve the table 1. We added the word "historical center" to highlight the place within the city where the survey was conducted and for the reader to identify the survey site.

Authors Dimension Indicators

[59]; [69];[10];[8]

 Satisfaction with the restaurant (SATREST) (SATREST1) Renown (known abroad)

(SATREST2) Tradition and roots in the local community

(SATREST3) Location

(SATREST4) Organization

(SATREST5) Power of attraction

(SATREST6) Food

(SATREST7) Service

(SATREST8) Atmosphere in the Restaurant

[8];

[40];

[96];[7]

 Gastronomy (GASTR)

 (GASTR1) Córdoba has exciting gastronomy 

(GASTR3) Córdoba is gastronomy

(GASTR4) Córdoba is cultural heritage

(GASTR5) Good value for money in restaurants

[40];[43]

Overall Image

(OVERALLIM) (OVERALLIM1) The overall image is positive

(OVERALLIM2) It was worth coming to Córdoba

(OVERALLIM3) It is a good place to visit

(OVERALLIM4) It has a good reputation

[46]; [47]; [39];[40]; [97]; [97]

Destiny loyalty (LOYDEST) (LOYDEST1) I will repeat my visit to this restaurant (Cordoba historical center)

(LOYDEST2) I will recommend that family and friends come to this restaurant (Cordoba historical center)

(LOYDEST3) I will visit Córdoba again

(LOYDEST4) I will recommend that family and friends visit Córdoba

RESPONSE TO REVIEWER 4

1. The main claims of the paper are very interesting and important for sustainable tourism development. The expansion of culinary tourism based on the traditional cuisine of a given country is perceived as a trend by UNWTO or Slow Food movement/organisation (which could be mentioned in the article).

Thank you very much for your suggestions. They have helped us improve the manuscript in several ways. Relevant changes in the manuscript are marked in red. We reviewed and references studies such as (see page 4, lines 14-16):

 “[… ] within the experience of visiting a destination. Thus, seek experience with local food, the unique flavors, customs or traditions, along with the quality of food, and local raw materials using concepts and movements such as the so-called Slow Food [28]."

[28] Çelebi, D. and S. GenÇ, Exploring the Slow Food Perception of Gastronomy and Culinary Arts Students. Journal of Tourism and Gastronomy Studies, 2021. 9(1): p. 99-110.

2. The introduction should draw more attention, describe the problem more clearly and provide rationale for the study. This part of the text seems chaotic, a few issues are repeated.

Thank you very much for your suggestions. They have helped us to improve the introduction chapter. It has been rewritten to provide clarity and justify the study. New paragraphs have been added and repeated questions have been removed (see page 2-5).

3. The style should be improved, the text contains many repetitions. The numbers instead of the authors' names also make a bad impression.

Thank you very much for your suggestions. They have helped us improve the manuscript in several ways. The entire text of the manuscript has been revised to eliminate repetitions. However, the numbers instead of the names in the citations has been kept due to the regulations of the journal.

4. The survey methodology should be described in more detail. The authors state that the research was conducted using a questionnaire and personal interview but the description of the results shows that only the survey data was taken into account. What was the interview about? Likewise, the description of the course of research is unclear.

Thank you very much for your suggestions. They have helped us improve the manuscript in several ways. Relevant changes in the manuscript are marked in red. Clarifications are included in" Methodology, Questionnaire and scales " (see page 15, lines 5-13):

“The restaurants where the visitor survey was conducted were located in the historic center of the city. This is the most touristic area of the city. The restaurants were selected for their location; secondly, for being recognized by professionals as examples of the traditional gastronomy of Cordoba and finally where the dishes had ingredients with the Protected Designation of Origin of the area.

The information collection procedure was carried out in December 2019 during the weekends, at mealtime, through a questionnaire where the pollster personally asked the tourist about his gastronomic experience. Questionnaire responses were collected by the pollster on the Tablet”.

5. The results were based on the responses of 139 respondents. It seems that this number is not sufficient to generalize the conclusions and apply them to populations other than tourists visiting traditional restaurants in Cordoba.

The results were based on 139 tourists who conducted a survey. Before conducting the survey he was asked if they had visited the city for reasons of traditional gastronomy. As explained in the text, the renowned restaurants offering traditional cuisine were selected and in the historic center of Cordoba. There are tourists who taste it by chance or just go to these restaurants and do not ask for this type of food, therefore, these tourists did not conduct the survey.

---

## [Editor Report · Decision Letter 1]

28 May 2021

The role of traditional restaurants in tourist destination loyalty

PONE-D-21-07254R1

Dear Authors,

We’re pleased to inform you that your manuscript has been judged scientifically suitable for publication and will be formally accepted for publication once it meets all outstanding technical requirements.

Kind regards,

Dejan Dragan, PhD

Academic Editor

PLOS ONE

Additional Editor Comments (optional):

The comments of all reviewers were appropriately considered. Accordingly, the acceptance of the paper is recommended. AE DD
---

## [Editor Report · Acceptance letter]

8 Jun 2021

PONE-D-21-07254R1 

The role of traditional restaurants in tourist destination loyalty 

Dear Dr. Hernández-Rojas:

I'm pleased to inform you that your manuscript has been deemed suitable for publication in PLOS ONE. Congratulations! Your manuscript is now with our production department. 

Kind regards, 

on behalf of

Dr. Dejan Dragan 

Academic Editor

PLOS ONE